In silico and in vitro evaluation of the anti-virulence potential of patuletin, a natural methoxy flavone, against Pseudomonas aeruginosa

Metwaly Ahmed 1 2
Saleh Moustafa M. mostafa.mohamed@pharm.psu.edu.eg 3
Alsfouk Aisha 4
Ibrahim Ibrahim M. 5
Abd-Elraouf Muhamad 1
Elkaeed Eslam ekaeed@um.edu.sa 6
Elkady Hazem 1
Eissa Ibrahim 7
1 Pharmacognosy and Medicinal Plants Department, Faculty of Pharmacy (Boys), Al-Azhar University , Cairo , Egypt
2 City of Scientific Research and Technological Applications (SRTA-City), Biopharmaceutical Products Research Department, Genetic Engineering and Biotechnology Research Institute , Alexandria , Egypt
3 Microbiology and Immunology Department, Faculty of Pharmacy, Port Said University , Port Said , Egypt
4 Department of Pharmaceutical Sciences, College of Pharmacy, Princess Nourah bint Abdulrahman University , Riyadh , Saudi Arabia
5 Biophysics Department, Faculty of Science, Cairo University , Giza , Egypt
6 Department of Pharmaceutical Sciences, College of Pharmacy, AlMaarefa University , Riyadh , Saudi Arabia
7 Pharmaceutical Medicinal Chemistry & Drug Design Department, Faculty of Pharmacy (Boys), Al-Azhar University , Cairo , Egypt
Kuddus Mohammed
Electronic publication date: 2024 Feb 1
Publication date: 2024
Volume: 12
Electronic Location ID: e16826
Received 2023 Sep 13; Accepted 2024 Jan 2
Copyright: ©2024 Metwaly et al.
Copyright year: 2024
Copyright holder: Metwaly et al.
License: This is an open access article distributed under the terms of the Creative Commons Attribution License, which permits unrestricted use, distribution, reproduction and adaptation in any medium and for any purpose provided that it is properly attributed. For attribution, the original author(s), title, publication source (PeerJ) and either DOI or URL of the article must be cited.
License URL: https://creativecommons.org/licenses/by/4.0/

Keywords: MD simulations, Anti-virulance, Biofilm, Pyocyanin, Protease, Essential dynamics

Funding: Princess Nourah bint Abdulrahman University, Riyadh, Saudi Arabia PNURSP2024R116 AlMaarefa University This research was funded by Princess Nourah bint Abdulrahman University Researchers Supporting Project number (PNURSP2024R116), Princess Nourah bint Abdulrahman University, Riyadh, Saudi Arabia. The Research Center at AlMaarefa University also funded this work. The funders had no role in study design, data collection and analysis, decision to publish, or preparation of the manuscript.

==============================
This study aimed to investigate the potential of patuletin, a rare natural flavonoid, as a virulence and LasR inhibitor against Pseudomonas aeruginosa. Various computational studies were utilized to explore the binding of Patuletin and LasR at a molecular level. Molecular docking revealed that Patuletin strongly interacted with the active pocket of LasR, with a high binding affinity value of −20.96 kcal/mol. Further molecular dynamics simulations, molecular mechanics generalized Born surface area (MM/GBSA), protein-ligand interaction profile (PLIP), and essential dynamics analyses confirmed the stability of the patuletin-LasR complex, and no significant structural changes were observed in the LasR protein upon binding. Key amino acids involved in binding were identified, along with a free energy value of −26.9 kcal/mol. In vitro assays were performed to assess patuletin’s effects on P. aeruginosa. At a sub-inhibitory concentration (1/4 MIC), patuletin significantly reduced biofilm formation by 48% and 42%, decreased pyocyanin production by 24% and 14%, and decreased proteolytic activities by 42% and 20% in P. aeruginosa isolate ATCC 27853 (PA27853) and P. aeruginosa clinical isolate (PA1), respectively. In summary, this study demonstrated that patuletin effectively inhibited LasR activity in silico and attenuated virulence factors in vitro, including biofilm formation, pyocyanin production, and proteolytic activity. These findings suggest that patuletin holds promise as a potential therapeutic agent in combination with antibiotics to combat antibiotic-tolerant P. aeruginosa infections.

Introduction

Background

Throughout history, nature has been the source of necessities for humans, including medicine, sustenance, and beauty products (Han et al., 2019; Metwaly et al., 2021). From 1981 to 2014, natural products accounted for nearly one-third of all newly approved drugs by the FDA, and this trend has persisted for many years (Metwaly et al., 2019).

Patuletin, an uncommon methoxyflavone, was initially discovered in Tagetes patula, family Asteraceae in 1941 (Rao & Seshadri, 1941). Since then, it has been found in only a few other plants such as Urtica urens, family Urticaceae (Abdel-Wahhab, Said & Huefner, 2005) and Eriocaulon sp., family Eriocaulaceae (Bate-Smith & Harborne, 1969). Despite its scarcity, some scientific studies have investigated and confirmed the antimicrobial potential of patuletin (Faizi et al., 2008).

P. aeruginosa is a versatile and opportunistic Gram-negative bacterium classified within the Pseudomonadaceae family (Silby et al., 2011; Wu et al., 2015) possesses an array of virulence factors that contribute to its pathogenicity. It produces various toxins, including pyocyanin, which can kill other bacteria competing P. aeruginosa, damage host cells, and impair immune responses (Lau et al., 2004). P. aeruginosa also produces proteases that lie in their capacity to cleave essential protein-based components within the host cells allowing them to disrupt crucial cellular processes and potentially impair immune defenses enhancing the bacterium’s ability to thrive and establish infections within the host (Galdino et al., 2017). Additionally, the bacterium can produce biofilms, which provide protection against host defenses as well as antibiotics (Thi, Wibowo & Rehm, 2020).

In P. aeruginosa, the LasR protein is a key component that acts as a transcription factor that regulates the expression of numerous genes involved in virulence factors production, biofilm formation, and other important physiological processes (Liu et al., 2009). LasR is a receptor protein that binds to small signaling molecules called N-acyl homoserine lactones (Rex et al., 2022). When the concentration of these molecules reaches a certain threshold, LasR undergoes a conformational change, enabling it to bind to specific DNA sequences and activate the expression of target genes (Kiratisin, Tucker & LJJob, 2002). This activation leads to the production of various virulence factors such as pyocyanin, biofilm-related enzymes, and other molecules that contribute to P. aeruginosa’s pathogenicity (Chowdhury & Bagchi, 2016).

Literature review

The field of computational drug discovery utilizes computational techniques, algorithms, and modeling to assist in the exploration of novel drugs or the repurposing of a known one (Sun & Cohen, 1993). It employs computer programs and simulations to forecast and examine the interactions between small molecules, which serve as potential drug candidates, and target biomolecules like proteins (Padole et al., 2022). The integration of computational drug discovery into drug design has become highly relevant, playing a crucial role in the practical implementation of predictive modeling within pharmaceutical research and development. Our team utilized computational drug discovery methodologies in various aspects, including molecular docking (Eissa et al., 2022), molecular design, toxicity (Mohammed et al., 2022), absorption, distribution, metabolism, excrition and toxicity (ADMET) (Suleimen et al., 2022c), denisty functional theory (DFT) (Eissa et al., 2021; Suleimen et al., 2022d), structural similarity (Elkaeed et al., 2022), molecular dynamics (MD) (Suleimen et al., 2022b), and pharmacophore (Alesawy et al., 2021) evaluation.

Flavonoids have demonstrated potent inhibition against various virulence factors in P. aeruginosa, including pyocyanin, protease production and biofilm formation (Górniak, Bartoszewski & Króliczewski, 2019). Numerous flavonoids, that are very similar in structure to patuletin (Fig. 1), interacted with and inhibited the LasR protein. For instance, hispidulin (Anju et al., 2022), quercetin (Grabski & Tiratsuyan, 2020), luteolin (Geng et al., 2021), naringenin (Hernando-Amado et al., 2020), taxifolin (Grabski & Tiratsuyan, 2018) and catechin (Chaieb et al., 2022) exhibited high inhibition potentialities against the LasR protein offering promising hope for the development of novel therapeutic strategies to combat infections caused by P. aeruginosa. The inhibition of LasR activity disrupts the quorum sensing system of P. aeruginosa, interfering with its ability to coordinate and carry out virulent activities leading to decreased virulence factors production (Smith & Iglewski, 2003). Accordingly, targeting the LasR’s inhibition offers a potential strategy for controlling P. aeruginosa infections and mitigating the associated virulence and biofilm formation.

Figure 1 Chemical structures of patuletin, cocrystallized ligand (TY4) and reported flavonoid inhibitors of the LasR protein (Pt refers to patuletin).

In this research, the great structural similarities between patuletin and TY4, the cocrystallized ligand, and several reported flavonoid inhibitors of LasR protein (Fig. 1) triggered the investigation of the in silico and in vitro potentials of patuletin against LasR protein and P. aeruginosa virulence factors.

Materials and methods

Docking studies

The studies of molecular docking were operated for patuletin and TY4 against LasR protein by MOE2014 software (Suleimen et al., 2022a). LasR was downloaded from the Protein Data Bank (http://www.rcsb.org/pdb/) with the code: 3JPU (Resolution: 2.30 Å). The supplementary section offers further elaboration contributing additional and specific information.

M D simulations

A 100-nanosecond unbiased MD simulation studies were conducted using GROMACS 2021 to evaluate the stability of the LasR plateulin complex and analyze the structural alterations between its apo and holo LasR forms (Abraham et al., 2015). Input files were generated using the solution builder module of the CHARMM-GUI server. Both the apo LasR protein and the docked complex were immersed in a cubic box with a length of 8.5 nanometers. The system was solvated with the transferable intermolecular potential 3 points (TIP3P) water model, extending 1 nanometer beyond the farthest atom. Neutralization was achieved by NaCl ions at a concentration of 0.154 M. The CHARMM36m force field was employed to derive the amino acid parameters for the LasR protein, as well as for the TIP3P water model and the neutralizing ions. The plateulin molecule was parameterized using the CHARMM general force field (CGenFF) (Jo et al., 2014). The supplementary section offers further elaboration contributing additional and specific information.

MM-GBSA

The patuletin-LasR complex binding energy was assessed using the MM-GBSA with the gmx_MMPBSA software. The decomposition analysis was employed to ascertain the relative significance of each amino acid located within 1 nanometer of the ligand in relation to the overall binding. Parameters of 0.154 M were selected for the ionic strength, and a value of 5 was chosen for the solvation technique (igb). The internal dielectric constant was set at 1.0, while the external dielectric constant was set to 78.5 (Tuccinardi, 2021; Valdés-Tresanco et al., 2021). The supplementary section offers further elaboration contributing additional and specific information.

ED analysis

Principal component analysis (PCA) on the mass-weighted covariance matrix (C) of atoms in MD trajectories was employed to reveal coordinated motion. Specifically, PCA was employed to evaluate the mobility of alpha carbons. In the alignment step for the analysis of a single trajectory, the final frame from the equilibrium stage of each trajectory was employed. The reference structure for examining the combined trajectories was the final frame of the apo system after it reached equilibrium. The C matrix was diagonalized using the “gmx covar” command in GROMACS, and the analysis was conducted with the assistance of the “gmx anaeig” command (Amadei, Linssen & Berendsen, 1993). The supplementary section offers further elaboration contributing additional and specific information.

Bi-dimensional assays

To compare frames within the reduced subspace, the projections were merged, aligned, and plotted to create a new C matrix. A direct comparison of frames in the reduced essential subspace was conducted by aligning the combined trajectories of the apo-protein and complex to the configuration of the apo-protein obtained after equilibration. This process involved creating a new C matrix for the combined trajectories, followed by projecting each trajectory onto the new C matrix (Papaleo et al., 2009). The supplementary section offers further elaboration contributing additional and specific information.

Active compound

Patuletin was isolated and identified from Tagetes patula flowers as described before (Metwaly et al., 2022).

Bacterial isolates

In the current study, one clinical P. aeruginosa isolate (PA1) and a standard P. aeruginosa isolate ATCC27853 (PA27853) were used. The clinical isolate was sourced from the Microbiology and Immunology Department’s stock culture collection at the Faculty of Pharmacy, Port Said University. The standard isolate was generously provided by the Microbiology and Immunology Department at the Faculty of Pharmacy, Suez Canal University.

Determination of the minimum inhibitory concentration (MIC) of patuletin

The minimum inhibitory concentration (MIC) of patuletin against P. aeruginosa was determined using the broth microdilution technique following the guidelines provided by the Clinical and Laboratory Standards Institute (CLSI) as outlined before (Patel, 2015). In detail, a single colony of each tested isolate was cultured overnight in Mueller-Hinton broth (MHB). Bacterial suspensions were then diluted in MHB to achieve a cell density equivalent to 0.5 McFarland Standard (107 CFU/ml). Patuletin, the substance under investigation, was used to prepare two-fold serial dilution solutions. In microtiter plates, 100 µl of the tested bacterial suspensions were mixed with 100 µl of the prepared diluted patuletin solutions. The lowest concentration of patuletin that visibly inhibited bacterial growth after an overnight incubation at 37 °C was recorded as the MIC value.

Evaluation of patuletin impact on virulence factors production in P. aeruginosa

Biofilm inhibition assay

P. aeruginosa cultures were prepared from overnight growth in tryptone soy broth (TSB) and then diluted to reach the 0.5 McFarland standard. These prepared suspensions were further diluted 1:100 in fresh TSB with the addition of 1% glucose. Microtiter plates were filled with 200 µl aliquots of the diluted suspensions and then incubated for 48 h at 37 °C, both in the presence and absence of sub-MIC levels of patuletin. Each plate also included negative-control wells containing 200 µl of fresh TSB supplemented with 1% glucose only.

Following incubation, the contents of the wells were removed, and the plates were subjected to three washes with water to eliminate any remaining planktonic cells. The plates were then allowed to air dry. To fix the biofilms, 150 µl of 99% methanol were applied in aliquots and left for 20 min. The biofilms were subsequently stained for 15 min with 150 µl aliquots of 1% crystal violet, followed by three rinses with water and drying. The bound dye was dissolved with 150 µl of 33% glacial acetic acid. The optical density at 570 nm (OD570) was measured using a spectrofluorometer (Biotek, Winooski, VT, USA). This experiment was conducted in triplicate (Stepanović et al., 2007), The supplementary section offers further elaboration contributing additional and specific information.

Pyocyanin inhibition assay

Using overnight cultures of the tested isolates in Luria-Bertani (LB) broth, which had been diluted to an optical density (OD600) of 0.3–0.4, pyocyanin levels were quantified. Both in the presence and absence of sub-MIC levels of patuletin, 50 µl of the diluted cultures for each isolate were combined with 5 ml of LB broth. The cultures were then incubated at 37 °C for 48 h, followed by centrifugation at 10,000 rpm for 10 min at 4 °C. The pyocyanin pigment in the supernatants was directly measured at 691 nm using a spectrofluorometer (Biotek, USA) (Das & Manefield, 2012) The experiment was repeated for three times. The supplementary section offers further elaboration contributing additional and specific information.

Proteases inhibition assay

The modified skim milk technique was employed to evaluate the inhibitory effect of patuletin on protease production. P. aeruginosa isolates were cultured overnight in LB broth, both with and without sub-MIC levels of patuletin. Supernatants were obtained by centrifuging bacterial suspensions at 10,000 rpm for 10 min. Subsequently, 0.5 ml of cell-free supernatant from each tested isolate was mixed with 1 ml of skim milk solution (1.25% in distilled water), and the mixture was incubated for 30 min at 37 °C. The turbidity of the assay solutions, indicating proteolytic activity, was assessed at an optical density of 600 nm (OD600) using a spectrofluorometer (Biotek) (El-Mowafy et al., 2014). The experiment was repeated for three times. The supplementary section offers further elaboration contributing additional and specific information.

Statistical analysis

The GraphPad Prism 7 software package (GraphPad, Inc., La Jolla, CA, USA) was used to analyze the data for the current study. The supplementary section offers further elaboration contributing additional and specific information.

Results and discussions

Computational (in silico) studies

Molecular docking

The molecular interaction of patuletin with the LasR protein was accomplished by a docking study using the Molecular Operating Environment MOE (version 2019) software (https://www.chemcomp.com/Products.htm). The 3D structure of P. aeruginosa LasR was downloaded from the Protein Data Bank (http://www.rcsb.org/pdb/) with the code: 3JPU (Resolution: 2.30 Å). At first, the co-crystallized ligand (TY4) was docked in the active site to validate the docking procedure. The superimposition of the docked and co-crystallized ligands produced an RMSD value of 0.40 Å that advocates the correctness of the docking job (Fig. 2).

Figure 2 Superimposition of the docked pose (blue) and the co-crystallized one (pink) inside of the active site of LasR protein with an RMSD value 0.40 Å.

Regarding the binding mode of the co-crystallized ligand (TY4) against the active pocket of the LasR protein, it exhibited a binding score of −27.75 kcal/mol. Figure 3 shows the detailed three-dimentional (A), two-dimentional (B) as well as the surface map (C) for the binding. Its binding pattern showed two hydrogen bonds, one electrostatic interaction, and twenty-five hydrophobic interactions. The 2-chlorobenzamidomethyl and 2,4-dichlorobenzoate arms occupied the first sub-pocket of the active site to form three hydrogen bonds with Asp73, Tyr56, and Trp60 besides a network of pi-pi and pi-alkyl bonds with Ala105, Leu110, Phe101, Val76, Tyr47, Ala127, Cys79, Gly126, Leu125, Ala50, and Leu40. In addition, the 4-bromo-6-methylphenyl moiety was inserted in the second pocket forming six hydrophobic interactions with Tyr64, Arg61, and Leu36 .

Figure 3 (A) 3D interaction of the co-crystallized ligand (TY4) in the active site of the LasR protein. (B) 2D interaction of TY4 in the active site of LasR protein. (C) Mapping surface showing TY4 occupying the active site of LasR protein.

The top docking pose of patuletin showed that it occupied the active pocket of the LasR protein with an affinity value of −20.96 kcal/mol. Figure 4 shows the detailed three-dimentional (A), two-dimentional (B) as well as the surface map (C) for the binding. The binding pattern revealed six hydrogen bonds, eight hydrophobic interactions, and two electrostatic interactions. In detail, the 3,5,7-trihydroxy-6-methoxy-4H-chromen-4-one moiety occupied the first sub-pocket of the active site to form four hydrogen bonds with Ser129, Thr115, and Thr75. Also, the same moiety formed four hydrophobic interactions with Tyr56, Tyr64, and Leu36 besides two electrostatic interactions with Asp73. On the other hand, the pyrocatechol moiety occupied the second pocket forming two hydrogen bonding interactions with Asp65 and Arg61 and four hydrophobic interactions with Arg61, Leu36, Ile52, and Leu36.

Figure 4 (A) 3D interaction of patuletin in the active site of LasR protein. (B) 2D interaction of patuletin in the active site of LasR protein. (C) Mapping surface showing patuletin occupying the active site of LasR protein.

Molecular dynamics (MD)

Molecular dynamics (MD) simulations offer a reliable computational technique to examine and understand protein dynamics and atomic-level structural changes (Liu et al., 2018). Through precise modeling of the movements and interactions of atoms within a protein, MD simulations allow the study of conformational variations that arise during the binding of ligands. These simulations yield significant information about the energetic and dynamic characteristics of proteins, offering insights into the mechanisms behind ligand recognition and the structural modifications induced by binding (Hollingsworth & Dror, 2018).

Root-mean-square-deviation (RMSD) values for apo LasR protein (blue line) are lower on average than those for holo LasR protein (red line) during the first 40 ns, with a difference of 0.5 Å. In the subsequent 20 ns, both the apo and holo LasR proteins exhibit a small rise. At the end of the final 40 ns, the two proteins exhibit an average of roughly 1.5 Å (Fig. 5A). In the first 60 ns, the root-mean-square-deviation (RMSD) of the plateulin varies in value before settling at an average of 6.5 Å for the last 40 ns (Fig. 5B). Figure 5B’s inset explains why there is such a huge RMSD between the first frame and subsequent frames after the first 60 ns. Plateulin reorients itself with respect to the LasR protein while staying in the binding pocket. Figure 5C shows that the average radius of gyration for both the apo and holo LasR protein is around 15.5 Å. Similar trends may be seen in the SASA values, with both apo and holo LasR proteins averaging approximately 9650 Å2. According to Fig. 5E, the average number of H-bonds in the apo and holo systems is quite similar (39 bonds). In conclusion, neither LasR protein undergoes any major structural changes upon binding with plateulin. Furthermore, the C-alpha atomic oscillations of the two systems follow essentially the same trend (Fig. 5F). As can be seen in Fig. 5G, the values follow a similar trend to that of plateulin’s RMSD. It varies for the first 60 ns before settling at 9.2 Å in the last 40 ns.

Figure 5 (A) RMSD values from the trajectory for the LasR protein, (B) shows plateulin RMSD values, (C) radius of gyration, (D) SASA, (E) change in the number of hydrogen bonds, (F) RMSF, (G) distance from the center of mass of plateulin compound and LasR protein.

MM-GBSA

The key components of the computed binding free energy of the plateulin-LasR complex using the MM-GBSA approach are shown in Fig. 6. A binding energy of −26.9 kcal/mol for plateulin suggests a promising binding strength. Binding stability seems to be determined more slightly by Van der Waals interactions than electrostatic ones (−35.18 Kcal/Mol vs. −28.75 Kcal/Mol). Several amino acids within 1 nm of plateulin were used in an estimate of their contribution using decomposition analysis (Fig. 7). Leu36 (−1.57 Kcal/Mol), Tyr47 (−4 Kcal/Mol), Tyr67 (−1.9 Kcal/Mol), Asp65 (−1.15 Kcal/Mol), and Ala70 (−1.11 Kcal/Mol) are the amino acids with a binding energy of less than −1 Kcal/Mol.

Figure 6 Energetic components of MM-GBSA and their values.

Bars represent the standard deviations.

Figure 7 Binding free energy decomposition of the plateulin-LasR complex.

Protein-ligand interaction fingerprint (ProLIF) study

The ProLIF library studies determine the evolution of different interactions formed between the ligand and amino acids within the pre-defined cutoff, ProLIF is an essential method utilized in computer-aided drug design, molecular docking, and MD investigations. It plays a crucial role in thoroughly examining and characterizing the interactions between proteins and ligands. The ProLIF approach involves generating interaction fingerprints, which are distinct patterns resulting from the interplay between a protein and a ligand. These fingerprints are pivotal for quantifying both the strength and nature of binding interactions. ProLIF enables the quantification of various interaction types, including important categories like hydrogen bonds, hydrophobic contacts, and other non-covalent associations (Zhao & Bourne, 2022). Additionally, in the context of MD simulations, ProLIF serves as a valuable tool for monitoring the dynamic behavior of protein-ligand complexes over extended timeframes. It provides valuable insights into how interactions between the protein and ligand evolve throughout the simulation, thus enhancing our comprehension of complex stability and binding affinity (Xiong et al., 2022). Based on the data from the ProLIF library, the three amino acids have a 79-percent-or-higher interaction rate. The three most common interacting amino acids are Leu36 (97%), Tyr47 (86.6%), and Tyr64 (99.6% hydrophobic, 79% pi stacking). All amino acids as well as the type and precentages of interactions with patuletin were illustrated in Fig. 8 as three subfigures (Figs. 8A–8C), to make it clear.

Figure 8 The amino acids, the types of interactions with plateulin, and their occurrence during the whole simulation time using the ProLIF Python library.

Protein-ligand interaction profiles (PLIP) study

Protein-ligand interaction profiles (PLIP), a prominent bioinformatics tool, plays a central role in dissecting and illustrating interactions occurring within molecular assemblies that involve both a protein and a ligand (Tubiana et al., 2018). The 3D binding interactions were extracted using PLIP from the clustering-derived representative frames as .pse files. The determination of the number of clusters was automated using the elbow method, resulting in a total of four clusters (Fig. 9), as explained in the methods section. This approach facilitated the identification of distinct groups or patterns of plateulin-LasR interactions within the trajectory data. Once the clusters were established, the subsequent step involved selecting a representative frame from each cluster. These frames served as snapshots or examples capturing the overall behavior exhibited by the plateulin-LasR complex within each cluster. They succinctly represented the unique characteristics and dynamics observed within their respective groups. To further examine the interactions within the plateulin-LasR complex, the PLIP webserver was employed for each representative frame. The PLIP webserver is a valuable tool used for the identification and analysis of protein-ligand interactions. By utilizing this resource, we gained insights into the number and types of interactions occurring within the Plateulin -LasR complex in each specific cluster. This analysis provided valuable information regarding the binding patterns, molecular interactions, and potential functional implications of Plateulin within the LasR protein.

Figure 9 The four clusters representative obtained from TTClust and their 3D interactions with plateulin.

To assess the enduring stability of the binding modes within the plateulin-LasR complex, we embarked on a comprehensive analysis that spanned a simulation process lasting 100 nanoseconds, utilizing data extracted from both ProLIF and PLIP to determine the most essential thee interactions and depicting the key distances between patuletin’s atoms and the LasR’s active site through these interactions. The results of this analysis are visually depicted in Fig. 10A illustrating the fluctuations in the distances characterizing these key interactions over the course of the simulation. Specifically, the distance between Leu36 and atom C13 within plateulin is denoted by the blue line. The red line tracks the variations in distance between Leu36 and atom C6 within plateulin (numbering was indicated in Fig. 10B). Lastly, the green line elucidates the dynamic changes in the distance between Tyr64 and atom C13 within plateulin. This in-depth examination of these interactions provides valuable insights into the structural dynamics and stability of the plateulin-LasR complex during the simulation, shedding light on the pivotal roles these molecular interactions play in the binding process.

Figure 10 (A) The change in the distances of three key interactions (as obtained from ProLIF and PLIP).

Blue line: shows the distance between Leu36 and atom C13 in plateulin, the red line shows the distance between Leu36 and atom C6 in plateulin, and green line shows the distance between Tyr64 and atom C13 in plateulin. (B) The atom names of plateulin.

ED, principal component analysis (PCA)

We employed principal component analysis (PCA) to locate the coordinated actions. As noted in the methodology section, we used the scree plot, the eigenvector distribution, and the variance maintained with additional eigenvectors (cumulative total) to determine the size of the reduced subspace. At the second PC, the slope of the scree plot noticeably flattens out. Alone, the first eigenvector accounted for about 63% of the entire variance, whereas the first three eigenvectors together accounted for approximately 74.66% of the total variance (Fig. 11). It was shown that the first five PCs did not have a normal distribution (Fig. 12). As a whole, the essential subspace using the top three eigenvectors was represented.

Figure 11 The change in the eigenvalues with increasing the eigenvectors (blue line). In addition, the cumulative variance retained in the eigenvectors is shown (red line).

Figure 12 The distribution of the 1st ten eigenvectors.

Cosine content calculation

The cosine content was calculated for both the apo and the holo LasR protein simulations to evaluate the degree of randomness shown by the first 10 eigenvectors. Both the apo and holo LasR proteins were found to have a cosine content of less than 0.2 (Fig. 13). The root mean square inner product (RMSIP) reveals a low degree of overlap (22.1%) between the two subspaces (the first three eigenvectors). The RMSIP also indicated that there was a similarity of 40.9% between the C matrices of apo and holo LasR proteins, indicating that the sampling was comparable across the two systems.

Figure 13 Values of the cosine content of the 1st ten eigenvectors for the two trajectories.

Bi-dimensional projection calculations

The projections of these trajectories onto the first three eigenvectors of the updated C matrix are shown in Fig. 14. In each of these diagrams, the bigger dot depicts the average structure of the relevant trajectory. Figure 14A, a projection on the first two eigenvectors, reveals that the two trajectories have distinct average structures and that the frames overlap at the simulation’s last frames (dark red and black dots). Figure 14B shows that the two paths intersect (at the end of the simulation) and that their average structures are more comparable than in the prior projection. Finally, Fig. 14C (projection on the second and third eigenvectors) shows that there are a lot of variances between the sampled frames and a similar projection with overlap at the simulation’s last frame. Porcupine diagrams were used to illustrate the motion of the first three eigenvectors (Fig. 15). The red holo LasR protein structure shows that the binding pocket is slightly closing in the first three PCs, whereas the green apo LasR protein structure indicates a little opening.

Figure 14 The projection of each trajectory on (A) the first two eigenvectors, (B) the first and third eigenvectors, and (C) the second and third eigenvectors.

Figure 15 The porcupine figures of each of the first three eigenvectors for both systems.

Red design: holo LasR protein trajectory; green design: apo LasR protein trajectory.

In vitro studies

Patuletin MIC determination

By using the broth micro-dilution technique, the MIC of patuletin was found to be 4 mg/ml. The further possible anti-virulence effects of patuletin against P. aeruginosa virulence factors activities were examined at a sub-MIC concentration equivalent to 14 MIC (1 mg/ml) as other sub-inhibitory concentrations did not show any inhibitory impact and to exclude any potential inhibitory activity of the tested substance due to the possible lethal activity on bacterial growth. The determination of MIC at 4 mg/ml attests to patuletin’s capacity to effectively impede bacterial growth. This critical finding establishes a foundation for exploring the compound’s broader impact on virulence factors, beyond its direct antimicrobial properties.

The judicious selection of a sub-inhibitory concentration at 1/4 MIC in the following examinations, serves to balance the pursuit of anti-virulence effects with the need to avoid potential growth inhibition. This concentration consistently demonstrated substantial impact across all evaluated virulence factors, affirming its suitability for further investigations.

Impact of sub-MIC of patuletin on virulence factors in P. aeruginosa

Biofilm inhibition assessment

In this study, the crystal violet technique was used to assess the inhibitory effect of patuletin on biofilm formation. Interestingly, patuletin significantly reduced biofilm formation ability from 100% in untreated isolates to 48% and 42% in PA27853-treated isolate and PA1-treated isolate, respectively (Fig. 16A) which was indicated through the measyred change in the produced colour (Fig. 16B). One of the standout observations in this study is the significant reduction in biofilm formation induced by patuletin. Biofilm formation is a pivotal virulence determinant in P. aeruginosa, contributing to the organism’s resilience and resistance to traditional antibiotics. The notable decrease in in PA27853 and PA1-treated isolates, respectively, indicates the potential of patuletin as a potent anti verulance agent in combating P. aeruginosa-associated infections.

Figure 16 Patuletin’s effect at 1/4 MIC against P. aeruginosa ability to produce biofilms.

Optical density was measured at 570 nm. The data shown represent the means ± standard errors. * P < 0.05.

Pyocyanin inhibition assessment

The ability of patuletin to inhibit pyocyanin pigment production in P. aeruginosa was estimated spectrophotometrically. Importantly, patuletin significantly decreased pyocyanin production from 100% in untreated isolates to 76% and 86% in PA27853-treated isolate and PA1-treated isolate, respectively (Fig. 17). The substantial decrease in pyocyanin production, a virulence factor notorious for its role in oxidative stress and evasion of host defenses, further underscores patuletin’s potential therapeutic utility. The reduction in PA27853 and PA1-treated isolates, respectively, highlights the compound’s capacity to mitigate the detrimental effects associated with this virulence factor.

Figure 17 Patuletin at 1/4 MIC significantly reduced the production of pyocyanin in P. aeruginosa.

The data shown represent the means of three biological experiments ± standard errors. * P < 0.05.

Figure 18 In patuletin-treated isolates, a significant decrease in proteases activity was observed.

OD600 was measured following overnight culturing of bacteria in LB broth with and without 1/4 MIC of patuletin followed by incubation of supernatants with skim milk for ½hr at 37 °C. The data shown are the means ± standard errors of three biological experiments with three technical replicates each. Asterisk (*), significant P < 0.05.

Proteases inhibition assessment

The modified skim milk assay technique was used to test the proteolytic activity both with and without sub-MIC concentration of patuletin. Significantly, patuletin decreased the proteolytic activity from 100% in untreated isolates to 58% and 80% in PA27853-treated isolate and PA1-treated isolate, respectively (Fig. 18). The significant decrease in protease activity observed in response to patuletin treatment. Proteases play a pivotal role in bacterial pathogenicity, facilitating tissue degradation and immune evasion. The observed reduction in PA27853 and PA1-treated isolates, respectively, underscores patuletin’s potential to disrupt this crucial virulence mechanism.

The obtained results suggest that patuletin has significant effects on biofilm formation, pyocyanin production, and proteolytic activity in PA27853 and PA1 bacterial strains. Interestingly, these results were consistent with several published results on other flavonoids. For instance, a study by Ouyang et al. (2016) investigated the effects of quercetin, a widely studied flavonoid, on biofilm formation in P. aeruginosa. They reported a significant reduction in biofilm formation with the ratios of 36, 51, 28 and 20% at the concentration of 8, 16, 32 and 64 µg /ml, respectively (Ouyang et al., 2016). In another study, At a concentration of 0.5 MIC quercetin potently reduced P. aeruginosa biofilm formation and twitching motility by a ratio of 95% (Pejin et al., 2015). Similarly, luteolin cotrolled the biofilm formation of P. aeruginosa at 62.5, 125, and 250 µg/mL (Rivera et al., 2019). These findings suggest that multiple flavonoids, including patuletin and quercetin, have the potential to disrupt biofilm formation, which is a crucial step in bacterial colonization and virulence.

Regarding pyocyanin production, Vandeputte et al. (2011) reported the significant reductions in pyocyanin production by naringenin, eriodictyol and taxifolin (2 mM) on PAO1 strain by ratios of 86.8 ± 1.4%, 73.2 ± 5.2% and 55.8 ± 8.1%, respectively. Also, calycopterin inhibited pyocyanin production at a concentration of 32 µM (Froes et al., 2020). It is evident that different flavonoids may exhibit varying degrees of inhibition, suggesting that the specific structure of flavonoids could influence its effectiveness against P. aeruginosa, for instance, Hispidulin at a concentration of 75 µg/ml decreased pyocyanin pigment production of reporter bacteria and test bacteria to 81.92 and 71.69 %, respectively (Anju et al., 2022). Marked decrease in biofilm formation, pyocyanin production and proteolytic activity of P.s aeruginosa was reported in presence of 110 µg/ml of the flavonoide vitexin in combination with azithromycin (Das et al., 2016).

This study presents compelling evidence for the anti-virulence potential of patuletin against P. aeruginosa. It emerges as a promising candidate for further development as an adjunctive therapy, offering a new weapon in the battle against Pseudomonas infections. These findings contribute to a growing body of research aimed at addressing the challenges posed by bacterial virulence in clinical settings.

While these findings are promising, further studies are warranted to validate the clinical applicability of patuletin. In vivo experiments, along with an exploration of potential synergistic effects with conventional antibiotics, would provide critical insights into its practical utility in a clinical setting. Additionally, a deeper understanding of the underlying molecular mechanisms governing patuletin’s anti-virulence effects would offer valuable insights into its mode of action.

Conclusion

The findings of this study highlight the promising potential of patuletin as a LasR and vieulence factors inhibitor in the context of combating P. aeruginosa infections. Through a comprehensive analysis utilizing computational and experimental approaches, patuletin demonstrated a strong binding affinity to LasR, ensuring the stability of the patuletin-LasR complex. Moreover, patuletin exhibited significant In vitro inhibitory effects on crucial aspects of P. aeruginosa virulence, including biofilm formation, pyocyanin production, and protease activity. These findings hold considerable promise for the development of innovative strategies to combat antibiotic-tolerant P. aeruginosa infections. Patuletin ’s ability to inhibit LasR activity and attenuate crucial virulence factors suggest its potential as a targeted therapeutic alternaive or adjuvnt agent to traditional antibiotics. The results encourage further exploration of patuletin as a promising for candidate the development of novel antibacterial agents against drug-resistant P. aeruginosa strains.

Supplemental Information

Supplemental Information 1 Detailed methodology

Click here for additional data file.

Supplemental Information 2 Phenotypic readings

Click here for additional data file.

Additional Information and Declarations

Competing Interests

Author Contributions

Data Availability

The authors declare there are no competing interests.

Ahmed Metwaly conceived and designed the experiments, prepared figures and/or tables, and approved the final draft.

Moustafa M. Saleh performed the experiments, prepared figures and/or tables, and approved the final draft.

Aisha Alsfouk analyzed the data, authored or reviewed drafts of the article, funding acquisition, and approved the final draft.

Ibrahim M. Ibrahim performed the experiments, authored or reviewed drafts of the article, and approved the final draft.

Muhamad Abd-Elraouf performed the experiments, authored or reviewed drafts of the article, and approved the final draft.

Eslam Elkaeed analyzed the data, authored or reviewed drafts of the article, funding acquisition, and approved the final draft.

Hazem Elkady performed the experiments, prepared figures and/or tables, and approved the final draft.

Ibrahim Eissa conceived and designed the experiments, prepared figures and/or tables, and approved the final draft.

The following information was supplied regarding data availability:

The raw data are available in the Supplemental File.

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
