# Peer review of "In silico and in vitro evaluation of the anti-virulence potential of patuletin, a natural methoxy flavone, against Pseudomonas aeruginosa"

_PeerJ, doi:10.7717/peerj.16826_

## Round 0.1 · original submission · Major Revisions

The manuscript titled 'In silico and In vitro evaluation of the anti-virulance potential of patuletin, a natural methoxyflavone, against Pseudomonas aeruginosa' is interesting and may be beneficial for further research in the field. However, the reviewers are not recommending it for publication in the present form due to some valid shortcomings in the manuscript. After critical evaluation of the manuscript, we would like to recommend it for major revision before publication. Some important comments that should be addressed in the revision are:
1. Improvement in scientific English language is needed.
2. The methodology section needs to be revised in detail and it should be reproducible.
3. Some figures have duplicate data (check reviewers' comments for detail) that should be corrected.
4. More detailed discussion is required as per the results obtained.
5. Check all corrections made in the annotated manuscript submitted by reviewers and address all the issues accordingly.

**Language Note:** The Academic Editor has identified that the English language must be improved. PeerJ can provide language editing services - please contact us at copyediting@peerj.com for pricing (be sure to provide your manuscript number and title). Alternatively, you should make your own arrangements to improve the language quality and provide details in your response letter. – PeerJ Staff
If the MS is revised successfully it may be considered for further review process.

Reviewer 1 ·

Basic reporting

The overall quality of this manuscript is acceptable but some problems should be addressed.

Experimental design

1. Abstract: It should be normally one paragraph, so no need to separate it into several paragraphs.
2. Keywords: These are key terms that did not appear in the manuscript title.
3. All abbreviations need full names when appearing for the first time including the abstract.

Validity of the findings

1. Line 161: How many replicates were conducted in each treatment?
2. Line 416: This is the conclusion section, so don't use "in conclusion" for the beginning.
3. Figures 17A and 18 have duplicated data.
4. Figures 16B and 17B are out of focus.

·

Basic reporting

The introduction is well written and presents relevant information for the work. I suggest that the authors emphasize the originality of this work. The experimental procedure is clear and presents in detail all the experiments carried out, in addition to being in accordance with the objectives proposed in the work. Regarding the results and discussion, I suggest that a more detailed discussion be carried out regarding the results obtained. The discussion seemed somewhat superficial to me. There are many issues in the discussion and need to be reviewed. Check all corrections made in the manuscript.

Experimental design

No comments.

Validity of the findings

No comments.

Additional comments

No comments.

·

Basic reporting

Title of manuscript Appropriate and within the scope of the journal
Abstract Sufficient but few grammatical errors Lines 34-35(Molecular docking revealed that Patuletin 35 strongly bound (bind) to the active pocket of LasR, with a high affinity value of -20.96 kcal/mol)
Line 45(Overall, this study demonstrated that Patuletin….), can use ‘in summary, generally,

Introduction 1. Too much literature on Pseudomonas aeruginosa and yet is not main content of study.-
a maximum of four lines/statements is sufficient. - the literature on flavonoids is recommendable (lines 99-110). however replace the word ‘near’ with ‘similar’(line 101 ---Numerous flavonoids, that are very near in structure to Patuletin)

2. paragraph rearranging _ The paragraph starting with line 92 could have been the first paragraph under ‘introduction’.

3. language - Covers Lines 61-74, 74 -91 Authors should avoid use of first-person pronouns (we, me etc). line 112, 136 260, 297, 314, (e.g line 112..us to examine the in silico and in vitro potential of Patuletin against LasR protein….)

Experimental design

- - methods not clear; well-designed but so summarised that replication could be a challenge.
- under materials and methods, procedures are not clear ; consider lines 122, 125, 125; the sentences are starting with the word ‘was’. The authors connected the sub-title with the next line make them one sentence. This is confusing.
- investagation perfomed was sufficent to draw conclusion on abilitty of the flavonoid to bind on the bateria

Validity of the findings

Publication • This is good contribution to research on flavonoids and their bioactivity
• data is accurate; methods used to obtain data are standard

• statistical analysis is sufficient to inform discussion and conclusions; the GraphPad Prism 7 software is very good for this kind of data.

• but is patuletin, the methoxtflavone, novel? did the Authors isolate it and from which plant of organism? And if yes, did they elucidate its structure?

Additional comments

manuscripts can be accepted: the concept is execellent; It is the arrangement of ideas and improvement of language that is needed.

---

## Round 0.2 · Minor Revisions

Thank you for revising the manuscript and addressing the comments of the reviewers. Howver, the revised manuscript need further consideration as per the comments from R3 (both in their text, and also in an attachment).

Kindly re-revised the MS and submit it for further evaluation.

·

Basic reporting

All requested changes were made by the authors, therefore I consider the work to be published.

Experimental design

No comments.

Validity of the findings

No comments.

Additional comments

No comments.

·

Basic reporting

Language and grammar - greatly improved after first review
Figures - clear, readable
Paragraphs / lines - Spacing is not uniform

Experimental design

Materials and methods The procedures are clear. however; -
(i) delete ‘we’ on line 164 under 2.5 Bio-dimensional Assays to read ‘To compare………… the reduced subspace, they were merged, aligned, a new C matrix created and projections plotted’.
(ii) line 182, broth microdilution technique – capitalise each word
(iii) In line 179, The sub-title 2.7. Determination--- should be in a stand-alone line as is the case for all titles and subtitles
(iv) line 183 ref. in brackets (Patel 2015) but sentence has ‘by patel et. al. (confusing). alternatively delete ‘as outlined by Patel et. al in 2015 and insert citation to avoid double referencing
(v) line 184, delete ‘to briefly summarise the procedure’. Give complete procedure for reproducibility
(vi) line 193 and 211, delete ‘The ability of P. aeruginosa to form biofilms was assessed as follows & The estimation of pyocyanin was performed as follows:’ this is not a step in the procedures

Validity of the findings

Statistical analysis - Satisfactory and sufficient to inform discussion and conclusion
Results and discussion - good
Conclusion Well drawn
References - more than enough. If the authors can be advised to use most recent and expunge relatively old references to avoid making the paper ‘heavy’ as a result of so many references. (between 20 – 30 references is recommendable

Additional comments

No comment

---

## Round 0.3 · accepted · Accept

Dear authors, as you have addressed all of the reviewers' comments successfully, your manuscript may be accepted for publication.